# Fairness in Federated Learning via Core-Stability

**Bhaskar Ray Chaudhury**   **Linyi Li**   **Mintong Kang**   **Bo Li**   **Ruta Mehta**
University of Illinois at Urbana Champaign

## Abstract

Federated learning provides an effective paradigm to jointly optimize a model benefited from rich distributed data while protecting data privacy. Nonetheless, the heterogeneity nature of distributed data, especially in the non-IID setting, makes it challenging to define and ensure fairness among local agents. For instance, it is intuitively "unfair" for agents with data of high quality to sacrifice their performance due to other agents with low quality data. Currently popular *egalitarian* and *weighted equity-based* fairness measures suffer from the aforementioned pitfall. In this work, we aim to formally represent this problem and address these fairness issues using concepts from co-operative game theory and social choice theory. We model the task of learning a shared predictor in the federated setting as a *fair public decision making* problem, and then define the notion of *core-stable fairness*: Given $N$ agents, there is no subset of agents $S$ that can benefit significantly by forming a coalition among themselves based on their utilities $U_N$ and $U_S$ (i.e., $\frac{|S|}{N} U_S \geq U_N$). Core-stable predictors are robust to low quality local data from some agents, and additionally they satisfy *Proportionality* (each agent gets at least $1/n$ fraction of the best utility that she can get from any predictor) and Pareto-optimality (there exists no model that can increase the utility of an agent without decreasing the utility of another), two well sought-after fairness and efficiency notions within social choice. We then propose an efficient federated learning protocol CoreFed to optimize a core stable predictor. CoreFed determines a core-stable predictor when the loss functions of the agents are convex. CoreFed also determines approximate core-stable predictors when the loss functions are not convex, like smooth neural networks. We further show the existence of core-stable predictors in more general settings using Kakutani's fixed point theorem. Finally, we empirically validate our analysis on two real-world datasets, and we show that CoreFed achieves higher core-stable fairness than FedAvg while maintaining similar accuracy.

## 1   Introduction

The success of many deployed machine learning (ML) systems crucially hinges on the availability of high-quality data. However, a single entity might not own all the data it needs to train the ML model it wants; instead, valuable data instances or features might be scattered in different organizations or entities. Distributed learning schemes such as federated learning (FL) [15] provide a training scheme that focuses on training a single ML model using all the data available in a cooperative way without moving the training data across the organizational or personal boundaries to protect data privacy.

On the other hand, given the heterogeneity in the local data distributions of different clients participating in FL, it has become quite challenging to design a classifier that performs reasonably across all of them. In fact, such an objective directly transfers to ensuring a fair performance of the classifier across all local data distributions. Therefore, fairness in FL has attracted substantial interest in the recent past [32, 22, 19, 13, 35].

In this work, we ask: *Is it possible to jointly optimize a centralized model with fairness guarantees regarding the heterogeneity of local agents? How to define such fairness such that no agents would*

36th Conference on Neural Information Processing Systems (NeurIPS 2022).

*intend to form an alternative coalition with a subset of agents? What could be the federated learning protocol that is able to ensure such fairness?*

To address the above research questions, we bring to bear notions from game theory and social choice theory. We first observe that federated learning can be cast into *public decision making*, where all agents derive their respective utilities from a common global decision, namely the globally learned model. Now the goal is to make this global decision fairly. One of the fundamental fairness measures in public decision making is that of core-stability [24]. Intuitively, we say that a set of agents can form a "blocking coalition" if each one of them can gain utility significantly (proportional to the size of their coalition) by training a unified model amongst themselves than the globally trained model. A globally trained model is core stable if there are no blocking coalitions.

We briefly justify the advantages of core-stability (a.k.a. core-stable fairness) over some of the existing notions of fairness in federated learning. Two commonly used fairness notions in federated learning are the *egalitarian fairness* [32, 22, 7, 35, 36, 25, 22] and *proportional fairness* [6, 5]. The egalitarian fairness aims to maximize the utility of the least happy agent[1]. In a proportional fairness, we want the ratios of the losses of any pair of the agents to be (super/ sub) proportional to the size of their respective datasets (this incentivizes the agents to share more of their data with the server). To avoid naming conflicts with our notion of *proportionality*, from here on, we refer to the proportional fairness introduced in [6] as *weighted equity based fairness* as this fairness compares the losses of every pair of agents. Usually, fairness notions that compare the utilities/ losses of agents with each other are called equity based fairness in social choice theory. We remark that both the notions are vulnerable if some agents have poor quality datasets. In particular, if one of the agents have high levels of noise in their data, call them noisy-agent, then their loss values will tend to be higher for most learnt predictors. The egalitarian fairness and the weighted equity based fairness may be unfair to the other agents as both may make decisions aiming to reduce the large loss incurred by the noisy-agent, thereby biasing the learning towards the data of the noisy-agent. A more desirable fairness property in this scenario maybe to compare the *loss percentage* of agents, i.e., the ratio of the loss incurred to the maximum loss that can be incurred, or equivalently *utility percentage* of agents, i.e., the ratio of the utility incurred to the maximum utility that can be incurred. Core-stability achieves this, together with other desirable properties (elaborated in Section 3).

**Our Contribution.** We formally define the *core-stable fairness* in federated learning by appropriately modeling agent's utility functions to capture their learning loss error. In particular, given a group of $N$ local agents, an aggregation protocol $P$, and an aggregated model $f$, we say that the model $f$ achieves core-stability if there are no coalition $S$ of agents that could benefit significantly by training a model with only their data (see Definition 1). Intuitively, this means that under a core-stable FL model, no agent has the incentive to deviate from current group and thereby obtain proportionally better aggregate utility from the final trained model. Additionally, we note that such a model $f$ will ensure sought-after guarantees of Proportionality (each agent gets $1/n$ times their best possible utility)[30] and Pareto-optimality (there is no predictor that can increase the utility of any agent without decreasing the utility of another agent) that equal-treatment based models [6, 32] may fail to.

Core-stability is a well-sought-after but a rare-to-exist notion. In case of *public goods* that resembles FL, existence of core-stable outcome was known only when agent's utility functions are linear [8]. While the utility functions that capture learning errors are inherently non-linear and highly complex making existing results inapplicable. We summarize our *contributions* as below.

- We formally extend core-stability from co-operative game theory to fairness in federated learning. We show that core-stability exists (in Section 4.1) as long as agent's utility functions are continuous with respect to the model parameters, and their non-negative conical combinations have a convex set of (local) optima. We prove this result using a fixed point formulation. In particular, we define a correspondence $\phi : P \to P$ on the set of all feasible predictors $P$, and ensure that any predictor $\theta^* \in P$ such that $\theta^* \in \phi(\theta^*)$ is core-stable. Thereafter we show that $\phi$ satisfies nice continuity like properties and therefore must admit a fixed point by Kakutani's fixed-point theorem [14].
- Next, we design an effective federated learning protocol CoreFed, which optimizes the final model by maximizing the protocol of agent's utilities. We prove that this protocol efficiently finds the core-stable model whenever the underlying utility functions are concave (see Section 4.2). Our protocol only needs gradient information from agents in each round.

---

[1]Equivalently maximize the minimum loss.

- We prove that above method directly applies to learning through linear regression or logistic regression, since their resulting utility functions are convex (see Section 4.3). For *Smooth Neural Nets (DNN)*, although the utility functions are (highly) non-convex, we manage to show that an approximate core-stable model can be learned within a local neighborhood (see Section 4.4).
- To capture cases where agents may have varying importance, we extend core-stability to *weighted* core-stability (in Section 4.5). We show that a *weighted* core-stable model is *weighted* proportional and Pareto-optimal, and that CoreFed protocol can be generalized to Weighted CoreFed to get the desired weighted guarantees.
- We conduct experiments on three datasets, and show that CoreFed achieves the core-stable fairness, while maintaining similar utility with the standard FedAvg protocol (see Section 5).

## 2 Related Work

**Fairness in Social Choice.** Fairness is one of the fundamental goals in many multi-agent settings. Over the years, motivated by applications, several notions of fairness have been proposed and investigated. Two fairness notions that are studied in many applications are that of *proportionality* [30] and *envy-freeness* [9]. Proportionality requires every agent to receive their proportional share of the best outcome, i.e., at least $1/n$ times their best possible utility. The notion of envy-freeness is defined in the context of resource allocation, where one aims to divide a set of items among agents fairly. In an envy-free allocation, no agent prefers the bundle of the other agent to her own. However, in *public decision making*, where all agents derive utility from a common global decision, this notion is not applicable! In public decision making, one of the most sought out fairness notion is that of *core-stability* [24]. Core-stability generalizes the notion of proportionality alongside other desirable properties like Pareto-optimality. The concepts of Core-stability find applications in many other settings in social choice and game theory and is well known to exist in some special settings [31] Another popular fairness notion is *equitability* which states that every agent should be equally happy, i.e., the utilities/ losses of all the agents should be the same. However, as explained in 1, using relaxations of equitability, may lead to undesirable outcomes if some agents have poor data quality. Over the last seven decades, several relaxations of envy-freeness [20, 26], proportionality [4, 23] and equitability [10] have been studied in computational social choice.

**Fairness in Federated Learning.** There have been several results on fairness in federated learning, each focusing on a particular aspect of the entire paradigm. For instance some work [12, 33] aim to establish fairness at the agent selection phase, where the server requests for updates from a selected subset of the agents. There are studies that aim to study fairness while training the global model such that it does not discriminate against protected groups [34] or that the model does not overfit the data of some agents at the expense of others [19, 22]. Earlier mentioned *egalitarian fairness* will fall under this category. Then, there are studies that consider fairness by evaluating the contribution of the agents towards training the joint model– for instance *weighted equity fairness* [6] does this based on the size of the data shared by the agents. Other studies assign significance to the agents based on Shapley values [29]. For a full taxanomy of fairness in federated learning, we urge the reader to check [28]. At large, most of the fairness notions are incomparable. As remarked in [6], "no one set of definitions is going to resolve the complex questions it raises".

## 3 Core-Stability in Federated Learning

**Problem Setup.** In any *predictive modelling* task, one would like to learn a function mapping from $\mathcal{X} \subseteq \mathbb{R}^d$ to $\mathcal{Y} \subseteq \mathbb{R}$. This includes both *regression* and binary *classification* where extension to multi-class classification is also feasible. We denote the space of such mappings as $\mathcal{F} = \{f_\theta \mid \theta \in P \subseteq \mathbb{R}^d\}$, where each $f_\theta \colon \mathbb{R}^d \to \mathbb{R}$ is a mapping function parameterized by the model weights vector $\theta$. Our goal is to determine $f_\theta \in \mathcal{F}$, such that for data sample $(x, y)$ drawn from the distribution $\mathcal{P}$, $f_\theta(x) \approx y$, i.e., $f_\theta(x)$ learns $y$ well. Since we identify a mapping function with each $\theta \in P$, we refer to $\theta$ as a *predictor* for the model [2].

**Utility Functions of the Agents.** The quality of a predictor $\theta$ is usually measured by the expected loss over the data distribution $\mathcal{P}$, i.e., $\mathbb{E}_{(x,y)\sim\mathcal{P}}\ell(f_\theta(x), y)$, where $\ell(\cdot, \cdot)$ is a prede-

---

[2]each $\theta$ is a predictor

fined loss function. Ideally, the training process would minimize this expected loss, i.e., attain $\theta^\star = \arg\min_{\theta \in P} \mathbb{E}_{(x,y) \sim \mathcal{P}} \ell(f_\theta(x), y)$. Since we are trying to determine a single predictor for several heterogeneous agents/ groups, we may not be able to give every group its best predictor. However, we want to choose the predictor that achieves *fairness* across all the groups. To define any notion of fairness from the classical economics literature, we need to define the **utility function** of a group for a predictor $\theta$. Intuitively, the utility is a measure of how good the predictor is for the group and its data. We define,

$$u(\theta) = M - \mathbb{E}_{(x,y) \sim \mathcal{P}} \ell(f_\theta(x), y) \tag{1}$$

where $M$ is a constant more than $(1 + \varepsilon)$ times the loss incurred from the worst predictor for agent $i$, i.e., $M \geq (1 + \varepsilon) \sup_{\theta \in P, (x,y) \in \mathcal{X} \times \mathcal{Y}} \ell(f_\theta(x), y)$. The scaling by $(1 + \varepsilon)$ is to avoid unnecessary degeneracies resulting from zero utilities, and we choose $\varepsilon \ll 10^{-5}$. Observe that the range of the utility function is from $0 < M\varepsilon$ to $M(1 + \varepsilon)$.

**Federated Learning and Fairness.** In the federated learning setting, we are given $n$ groups. Each group $i$ has their loss function $\ell_i()$ and correspondingly a utility function $u_i()$ for each choice of a predictor $\theta \in P$. We now define the fairness criterion. Recall that given $n$ groups, our goal is to choose a $\theta \in P$, such that we are fair to all the involved agents. The fairness notion here is *core-stability*.

**Definition 1** (Core-Stability). A predictor $\theta \in P$, is called core stable if there exists no other $\theta' \in P$, and no subset $S$ of agents, such that $\frac{|S|}{n} u_i(\theta') \geq u_i(\theta)$ for all $i \in S$, with at least one strict inequality.

Intuitively, core-stability implies that there is no subset of agents that can benefit "significantly" by forming a coalition among themselves, i.e., if we were to choose any other $\theta' \in P$ only considering the utility functions of the agents in $S \subseteq n$, then there is some agent who's (multiplicative) gain in utility will be strictly less than a factor $n/|S|$, i.e., there is no substantial benefit for this agent if she chooses to belong to the set $S$. Furthermore, *core-stability* gives some classical fairness and welfare guarantees. In particular, note that every agent $i$ gets at least $1/n$ times her best utility, i.e., the utility derived from the best possible mapping for agent $i$. Mathematically $u_i(\theta) \geq 1/n \cdot u_i(\theta')$ for all $\theta' \in P$ (setting $S = \{i\}$ in Definition 1). This fairness is called *Proportionality* [30]. Formally,

**Definition 2** (Proportionality). A predictor $\theta \in P$ is proportional if and only if for all $\theta' \in P$, we have $u_i(\theta) \geq \frac{u_i(\theta')}{n}$ for all $i \in [n]$.

Similarly, observe that there exists no predictor $\theta' \in P$ where $\sum_{i \in [n]} \frac{u_i(\theta')}{u_i(\theta)} > n$ (setting $S = [n]$ in Definition 1). This implies that there is no predictor that can increase the utility of some agent without decreasing the utility of another. We call this property *Pareto-optimality*. Formally,

**Definition 3** (Pareto-optimality). A predictor $\theta \in P$ is Pareto-optimal if and only if there exists no other $\theta' \in P$, such that $u_i(\theta') \geq u_i(\theta)$ for all $i \in [n]$ with at least one strict inequality.

Core is a central concept within cooperative game theory, defined to capture "no deviating sub-group" property and is considered very strong. However, it is well known to exist only in special cases [31]. We now elaborate the advantages of core-stability over some of the existing fairness concepts in federated learning.

## 3.1 Advantages of Core-Stability

As briefly mentioned in the introduction, core-stability is robust to low local data qualities of some agents, unlike the FedAvg or federated learning based on egalitarian or weighted equity based fairness. We elaborate this with a small example: consider three agents that contribute equal amount of data, and say agent 3 has poor data quality, i.e., there exists no proper predictor for agent 3's data, or equivalently for all predictors $\theta \in P$, the loss function of this agent is very high (and utility is very low). Concretely, consider two predictors $\theta_1$ and $\theta_2$. Under $\theta_1$, agents 1 and 2 incur a loss of $a$ and agent 3 incurs a loss of $M \cdot a$ (think of $M$ as a very large integer). Now, under $\theta_2$, agents 1 and 2 have a loss of $10a$ and agent 3 has a loss of $0.9Ma$. Observe that $\theta_2$ is preferable over $\theta_1$ under egalitarian fairness (as $0.9Ma \gg 10a$) and also by FedAvg as it has a lower total average loss ($0.1Ma \gg 9a$). Similarly, a learning algorithm based on weighted equity fairness would prefer $\theta_2$, as ratio of the losses between agents 1 (or 2) and 3 is significantly high in both $\theta_1$ and $\theta_2$ and is lesser in $\theta_2$. However, intuitively, $\theta_1$ seems fairer, as agent 3 is not substantially worse off in $\theta_1$ than it is in

$\theta_2$ (by a factor 1.1), while agents 1 and 2 are significantly better off in $\theta_1$ (by a factor 10). Note that in this example $\theta_2$ is not a core-stable predictor, as agents 1 and 2 have an incentive to break off and improve substantially (intuitively $\theta_2$ is very unfair to them). We say that core-stable predictors are robust to low data quality of specific agents, as we never compare the losses of two agents with each other; rather our comparison is more along the lines of *loss percentages*, i.e., the ratio of the loss to the maximum possible loss incurred by the agent.

The robustness to poor local data quality of agents of core-stable predictors is a parallel to the property of *scale-invariance* that core-stable allocations exhibit in social choice theory. In particular, scaling the utility of any single agent does not alter the core-stable allocation. Similarly, Egalitarian, utilitarian[3] and equity based fairness suffer from being responsive to scaling [3].

## 4  Core-Stability in Federated Learning

In this section, we prove the existence of core-stability under certain assumptions on the loss functions of the individual agents/ groups (Section 4.1). Then, in Section 4.2, we give a distributive training protocol CoreFed to determine a core-stable predictor when the loss functions are convex[4]. Finally, by applying the theory developed in Sections 4.1 and 4.2, we show that CoreFed determines a core stable predictor in Linear Regression, and in Classification with Logistic Regression (Section 4.3). Finally, in Section 4.4 we show that CoreFed determines an approximate core stable predictor in Deep Neural Networks.

### 4.1  Existence of Core-Stability in Federated Learning

We show that core stable predictors exist in the federated setting if the utility functions of the agents satisfy the following conditions:

1. The utility function of each agent is continuous.

2. The set of maximizers of any conical combination of the utility functions is convex i.e., for all $\langle \alpha_1, \alpha_2, \ldots, \alpha_n \rangle \in \mathbb{R}^n_{\geq 0}$, the set $C = \{\theta \mid \sum_{i \in [n]} \alpha_i u_i(\theta) \text{ is maximum }\}$ is convex.

To the best of our knowledge, prior to this work, the existence of core-stability in public fair division was shown only for linear utility functions by [8]. Utility functions that satisfy the above two conditions cover several other utility functions and is therefore a strict generalization of linear utility functions. We show the existence of core-stability for instances satisfying conditions 1 and 2 above using Kakutani's fixed point theorem. For completeness, we state the Kakutani's fixed point theorem.

**Definition 4.** [Kakutani's Fixed Point Theorem] A *correspondence* or equivalently a *set valued function* $\phi \colon D \to 2^D$ admits a fixed point, .i.e., there exists a point $d \in D$, such that $d \in \phi(d)$, if

1. $D$ is non-empty, compact, and convex.

2. For all $d \in D$, $\phi(d)$ is non-empty, convex and compact.

3. $\phi()$ has a closed graph, i.e., for all sequences $(d_i)_{i \in \mathbb{N}}$ converging to $d^*$ and $(e_i)_{i \in \mathbb{N}}$ converging to $e^*$, such that $d_i \in D$ and $e_i \in \phi(d_i)$, we have $e^* \in \phi(d^*)$.

We define a correspondence or a set valued function $\phi \colon P \to 2^P$ where $P$ is the set of all feasible predictors. In particular, for all $\theta \in P$, we set

$$\phi(\theta) = \left\{ \{d \mid \sum_{i \in [n]} \frac{u_i(d)}{u_i(\theta)} \text{ is maximum }\} \right.$$

We first observe that any fixed point of $\phi$ corresponds to a core stable classifier.

**Lemma 4.1.** *Let $\theta \in P$ be such that $\theta \in \phi(\theta)$. Then, $\theta$ is a core-stable predictor.*

The proof of Lemma 4.1 can be found in the Appendix. Now, it suffices to show that $\phi$ admits a fixed point. In particular, note that the domain of $\phi$, $P$ is non-empty, compact, and convex. Similarly, for

---

[3]This is the parallel to FedAvg in social choice theory.

[4]The assumptions made in Section 4.1 to show only existence of core-stability are weaker than the convexity assumptions in Section 4.2.

every $\theta \in P$, $\phi(\theta)$ is non-empty, compact, and convex. By Kakutani's fixed point theorem, it only remains to show that $\phi()$ has a closed graph, to ensure that $\phi()$ admits a fixed point.

**Lemma 4.2.** *The correspondence $\phi()$ has a closed graph.*

The detailed proof can be found in the Appendix. Intuitively, since the utility functions are continuous and non-zero, the optima of $\sum_{i \in [n]} \frac{u_i(d)}{u_i(\theta)}$ over $d \in P$, also changes continuously with $\theta$. We are ready to prove the main result of this section.

**Theorem 1.** *In any federated learning setting, where the agent's utilities are continuous and the set of maximizers of any conical combination of the agents utilities is convex, a core-stable predictor exists.*

*Proof Sketch.* Any fixed point of $\phi()$ corresponds to core stable predictor (Lemma 4.1). It suffices to show that $\phi()$ admits a fixed point under assumptions in Theorem 1. To this end, note that domain $P$ of $\phi()$ is non-empty, compact, and convex. For all $\theta \in P$, $\phi(\theta)$ is convex by assumption in Theorem 1. Finally $\phi()$ has a closed graph by Lemma 4.2, and thus $\phi()$ admits a fixed point.

**Implications.** Theorem 1 describes the conditions under which core-stable predictors exist. We briefly state how to adapt the proofs to show the existence of *locally core-stable* predictors for more general utility functions. Lemmas 4.1 4.2, and Theorem 1 are valid even if we change the definition of $\phi(\theta)$ to the set of maximizers of $\sum_{i \in [n]} u_i(d)/u_i(\theta)$ over $d \in \mathcal{B}(\theta, r)$ (instead of $d \in P$), i.e., we define $\phi(\theta)$ to be the set of maximizers in the local neighbourhood of $\theta$ (within distance $r$ to $\theta$). In such a case, we only need conditions 1 and 2 to be true within a radius of $r$ of every point, i.e., within $\mathcal{B}(\theta, r)$ for all $\theta \in P$. These guarantees typically tend to be true for small values of $r$ in Deep Neural Networks. Thus, the predictor corresponding to the fixed point of $\phi$ will satisfy core-stability when restricted to predictors within distance $r$ to it, i.e., it is a locally core-stable predictor.

## 4.2 Computation of a Core-Stable Predictor When Utility Functions are Concave

In this section, we show that under certain assumptions on the utility functions, we can describe an efficient distributed protocol that computes the core-stable predictor. In particular, we look into the scenario, where the utility function of each agent is concave. Note that this would automatically satisfy the conditions in Theorem 1, as any conical combination of concave functions is also concave and will admit a convex set of maximizers.

We first show that a core stable predictor can be expressed as an optima of a convex program. In particular, any predictor that maximizes the product of utilities of the agents, i.e., $argmax_{\theta \in P} \prod_{i \in [n]} u_i(\theta)$ (or equivalently the sum of logarithms of the utilities of the agents), is core stable.

$$\begin{aligned} \text{maximize} \quad & \mathcal{L}(\theta) = \sum_{i \in [n]} \log(u_i(\theta)) \\ \text{subject to} \quad & \theta \in P \end{aligned} \tag{2}$$

Observe that if the utility of each agent is concave, then the above program is convex. Since the logarithm is a concave increasing function, each $\log(u_i())$ is a concave in $\theta$ and the sum of concave functions is concave. Thus, 2 is a concave maximization subject to convex constraints.

**Theorem 2.** *If $u_i()$ is concave for all $i \in [n]$, then any predictor $\theta^*$ that maximizes the convex program 2 is core-stable.*

*Proof Sketch.* We defer a formal proof to Appendix C. The main technical ingredient of our proof is to show that if $\theta^*$ is a solution to the convex program 2, then, for any other predictor $\theta' \in P$, we have $\sum_{i \in [n]} \frac{u_i(\theta')}{u_i(\theta^*)} \leq n$. Now if $\theta^*$ is not core-stable, then there exists an $S \subseteq [n]$ and $\theta' \in P$, such that $u_i(\theta') \geq n/|S|u_i(\theta^*)$ for all $i \in S$ with at least one strict inequality, then we have $\sum_{i \in [n]} \frac{u_i(\theta)}{u_i(\theta')} \geq \sum_{i \in S} \frac{u_i(\theta)}{u_i(\theta')} > n/|S| \cdot |S| = n$, which is a contradiction.

**Implications.** The proof of Theorem 2 shows that the strong utilitarian property of $\sum_{i \in [n]} \frac{u_i(\theta')}{u_i(\theta^*)} \leq n$ for any $\theta' \in P$ implies core-stability of $\theta^*$ under *any* (non-negative) utility functions. Clearly, such a $\theta^*$ must be Pareto-optimal, and furthermore, the inequality implies that at $\theta'$ computed by any other classical method, if some agents gain, then some other agents must be loosing by a lot. Secondly,

under concave utilities optima of convex program equation 2 satisfies this property, and hence can be computed in efficiently. Below we discuss a distributed protocol for the same.

**Tightness of Our Guarantees.** We make a remark that there are instances, where the guarantees provided by core-stability is tight. These instances typically exhibit large heterogeneous behaviour in their training data. Our guarantees work under the assumption that the utility functions are concave and the domain of predictor is a convex set. Consider the following scenario: Our goal is to choose a predictor $\theta \in \mathcal{C}$, where $\mathcal{C} = \{c \in \mathbb{R}^n_{\geq 0} \mid \sum_{i \in [n]} c_i = 1\}$ (so $\mathcal{C}$ is a convex set). Now, let there be $n$ agents, and $u_i(c) = c_i$ for all $i \in [n]$, and $c \in \mathcal{C}$ (utility functions are linear and therefore concave). Intuitively, each agent has their ideal predictor to be a distinct axis aligned hyperplane (capturing the heterogeneity in data). Observe that for each agent , the best possible utility is 1, as there exists a predictor $c^*$ such that $c_i^* = 1$ and $c_k^* = 0$ for all $k \neq i$. Now, note that, for any predictor $c \in \mathcal{C}$, we have $\sum_{i \in [n]} c_i = 1$. Thus, for any classifier $c$ chosen, there exists an agent $i$, such that $u_i(c) = c_i \leq 1/n$. Thus, for each predictor $c$, there exists a set $S = \{i\}$ and a another predictor $c^* \in \mathcal{C}$ such that $\sum_{i \in S} u_i(c^*)/u_i(c) > n$. Observe that even our guarantees of proportionality are tight in this example. This example shows that the scaling factor of $|S|/n$ is unavoidable.

We now propose a distributed SGD framework to determine a core stable predictor. We call our Algorithm as CoreFed (Fully outlined in Algorithm 1 in the appendix).

**CoreFed.** Observe that for convex losses, we can directly solve this maximization to the optimal to achieve core-stability. For non-convex losses such as those for DNNs, we apply gradient descent to maximize the objective. Suppose we are training on $n$ finite samples $\{(x_i, y_i)\}_{i \in [n]}$ drawn from the data distribution $\mathcal{P}$, which constitute empirical distribution $\hat{\mathcal{P}}_n$. We observe that, the gradient can be expressed as an conical combination of the gradients of each group:

$$\nabla_\theta \mathcal{L}(\theta) = \sum_{i \in [n]} \frac{\nabla_\theta u_i(\theta)}{u_i(\theta)} = \sum_{i \in [n]} \frac{-\nabla_\theta \mathbb{E}_{(x,y) \sim \hat{\mathcal{P}}_n^{(i)}} \ell(f_\theta(x), y)}{M_i - \mathbb{E}_{(x,y) \sim \hat{\mathcal{P}}_n^{(i)}} \ell(f_\theta(x), y)}. \tag{3}$$

Therefore, for each group, we reweight its gradients or weight updates by $(M_i - \mathbb{E}_{(x,y) \sim \hat{\mathcal{P}}_n^{(i)}} \ell(f_\theta(x), y))^{-1}$ and then sum up to get the final weight update in each iteration, which leads to a distributed training protocol shown in Algorithm 1.

This protocol is similar to standard FedAvg. However, in CoreFed the model weight updates are weighted then aggregated at each iteration, while in FedAvg, model weights are directly averaged and aggregated at each iteration. In the limit that each local update uses single step with entire dataset, $\Delta \theta_s = -\eta \frac{1}{|\mathcal{D}_s|} \sum_{i=1}^{|\mathcal{D}_s|} \nabla_{\theta^t} \ell(f_{\theta^t}(x_s^{(i)}), y_s^{(i)})$, where $\mathcal{D}_s = \{(x_s^{(i)}, y_s^{(i)}) : 1 \leq i \leq |\mathcal{D}_s|\}$ is the training dataset on device $s$. Therefore, the global update is a unbias gradient descent step of the objective $\sum_s \log(M_s - \frac{1}{|\mathcal{D}_s|} \sum_{i=1}^{|\mathcal{D}_s|} \ell(f_{\theta^t}(x_s^{(i)}), y_s^{(i)})) = \mathcal{L}(\theta_t)$ where $\mathcal{L}(\cdot)$ is defined in 2.

### 4.3 Core-Stability in Linear Regression and Classification with Logistic Regression

We now discuss some of the predictive models, where the concavity requirements of the utility function is satisifed. Note that a necessary and sufficient condition for $u_i()$ to be concave is that the loss function $\ell()$ should be convex in $c$. Here we elaborate few scenarios where this is true. Suppose we are training on $n$ finite samples $\{(x_i, y_i)\}_{i \in [n]}$ drawn from the data distribution $\mathcal{P}$, which constitute empirical distribution $\hat{\mathcal{P}}_n$.

**Linear Regression.** These are the scenarios where we map our input variables to a real number (not discrete class labels). In this case, we have $f_\theta(x) = \theta^\mathsf{T} x$. Observe that the *regression loss*, then $\mathbb{E}_{(x,y) \sim \hat{\mathcal{P}}_n} \ell(f_\theta(x), y) = \mathbb{E}_{(x,y) \sim \hat{\mathcal{P}}_n} (\theta^\mathsf{T} x - y)^2$ would be convex in $\theta$.

**Lemma 4.3.** *CoreFed determines a core-stable predictor in a federated learning setting training linear regression.*

**Classification with Logistic Regression.** In classification tasks we map the input variables to discrete class labels. A commonly used loss function in classification is logistic regression. Given a classifier $\theta$ and a scalar $c \in \mathbb{R}$, an agent $i$'s loss is given by $\ell_i(\theta, c) = \frac{||\theta||_2}{2} + \alpha \cdot \sum_{i \in [n]} \log(e^{-y_i(\theta^\top x_i + c)} + 1)$ [27, 1, 11]. It is well known that $\ell_i(\theta, c)$ is convex (see, e.g., [11]). Thus, $u_i(\theta, c) = M_i - \ell_i(\theta, c)$ where $M_i = \arg\max_{\theta \in P, c \in \mathbb{R}} (\ell_i(\theta, c))$, is concave.

**Lemma 4.4.** *CoreFed determines a core-stable predictor in a federated learning setting training classification with logistic regression.*

### 4.4 Approximate Core-Stability in Deep Neural Networks

Theorem 2 requires that $u_i(\theta)$ is concave in terms of $\theta$ and global optimality for the objective $\mathcal{L}(\theta) = \prod_{i \in [n]} u_i(\theta)$ to achieve core-stability. However, these two conditions are challenging to be satisfied for DNNs, where the training loss is non-convex and common training methods, which are based on first-order gradients, are not guaranteed to absolutely converge. In this more general scenario, the following theorem shows the relaxed local core-stability that we can attain for approximately first-order converged predictor (i.e., predictor with small local gradient $||\nabla_\theta \mathcal{L}(\theta)||_2 \leq \epsilon$).

**Definition 5.** A predictor $\theta \in P$, is called $(d, k)$-pseudo core stable, where $d > 0, k > 1$ if there exists no other $\theta' \in P$ such that $||\theta' - \theta||_2 < d$, and no subset $S$ of agents, such that $\frac{|S|}{kn} u_i(\theta') \geq u_i(\theta)$ for all $i \in S$, with at least one strict inequality.

**Theorem 3.** *For all $i \in [n]$, if $u_i(\theta)$ is $\beta$-smooth in terms of $\theta$ within $\{\theta' : ||\theta - \theta'||_2 \leq d\}$, and $||\nabla_\theta \mathcal{L}(\theta)||_2 \leq \epsilon$, then $\theta$ is a $(d, k)$-pseudo core stable predictor, where*

$$d = \frac{-\epsilon + \sqrt{\epsilon^2 + 2\beta(k-1)n \sum_{i \in [n]} u_i(\theta)^{-1}}}{\beta \sum_{i \in [n]} u_i(\theta)^{-1}}. \tag{4}$$

**Implications.** We defer the proof to Appendix E. Theorem 3 states that, for smooth neural networks, there exists no predictor $\theta'$ in the neighborhood with $\ell_2$ radius $d$, that any subset of agents prefer "significantly". Although our guarantees are local guarantees, we remark that global fairness guarantees are unlikely for DNNs. Most of the fairness guarantees in computational social choice and game theory crucially require the agents to have convex preferences, i.e., the level sets of the utility functions need to be convex. There are impossibility results for fairness when the agent's preferences are non-convex. However, while non-convex consumer preferences are not interesting from an economic standpoint, our current work finds an application for these preferences in fairness in federated learning with DNNs.

### 4.5 Weighted Core-Stability

In this section, we show how to generalize all of our results (Theorems 1, 2, and 3) when we want to train the joint predictor to fit the data of certain agents more than some others. In particular, for each agent $i$, if we assign weight $w_i$, indicating the desired bias of the final trained model towards agent $i$[5], then with subtle modifications, we can show the existence of a *weighted core stable* predictor, when the utility functions of the agents satisfy the conditions in Theorem 1. Formally,

**Definition 6** (Weighted Core-Stability). Given the weight vector $w = \langle w_1, w_2, \ldots, w_n \rangle$, a predictor $\theta \in P$, is weighted core-stable if and only if there exists no other predictor $\theta' \in P$ and a subset of agents $S \subseteq [n]$ such that $\frac{\sum_{j \in S} w_j}{\sum_{j \in [n]} w_j} \cdot u_i(\theta') \geq u_i(\theta)$ for all $i \in S$ with at least one strict inequality.

Note that, when all the agents have the same weight, *e.g.,* $w_i = 1, \forall i \in [n]$, then weighted core-stability matches core-stability. At a high-level the concept is the same, no group of agents can significantly benefit by forming a coalition within themselves. However "significantly" means a multiplicative increase by $\frac{\sum_{j \in [n]} w_j}{\sum_{j \in S} w_j}$ (instead of $|S|/n$ for the unweighted case), i.e., it is dependent on the total weight of the set $S$. We make the aforementioned guarantee more intuitive by considering special cases of $S$. When $S = \{i\}$, our guarantees say that agent $i$ gets a utility of $w_i/(\sum_{j \in [n]} w_j)$

---

[5]Following the light of [6], one possible candidate can be $w_i = \mathcal{D}_i$, i.e., set $w_i$ to the size of the data shared by agent $i$ with the model.

fraction of her maximum utility, i.e., the utility of agents with higher weights are prioritized. We call this *weighted proportionality*. Also note that by setting $S = [n]$, we get Pareto-optimality (similar to the unweighted case).

Furthermore, by changing the convex program 2 to maximizing $\sum_{j \in [n]} w_j \log(u_j(\theta))$ instead of $\sum_{j \in [n]} \log(u_j(\theta))$, we can get the weighted version of Theorem 2. This also suggests a simple generalization of CoreFed to Weighted-CoreFed and all our extensions in Sections 4.3 and 4.4 will also generalize to the weighted setting.

## 5 Empirical Evaluation

We evaluate our fair ML method CoreFed and baseline FedAvg [21] on three datasets (Adult, MNIST and CIFAR-10) on linear model and deep neural networks. We show that the model trained with CoreFed is able to achieve core-stable fairness, while maintaining similar utility with the standard FedAvg protocol, which cannot guarantee to achieve core-stable fairness.

### 5.1 Experiment Setup

**Dataset.** We evaluate our algorithm CoreFed on Adult [2], MNIST [17] and CIFAR-10 [16] datasets. To perform federated learning on heterogeneous data, we construct the non-IID data by sampling the proportion of each label from Dirichlet distribution for every agent, following the literature [18].

**Models.** We train a logistic regression classifier on Adult data. We use a CNN, which has two 5x5 convolution layers followed by 2x2 max pooling and two fully connected layer with ReLU activation for MNIST and CIFAR-10. We also evaluate with a more complex network VGG-11 on CIFAR-10. For Adult dataset, the utitility is selected as $M - \ell_{log}$ where $\ell_{log}$ is the logistic loss. For CIFAR-10 and MNIST, we use cross entropy loss $\ell_{ce}$ as the training loss with utility $U$ becomes $M - \ell_{ce}$. $M$ is set to be 3.0, 1.0 and 3.0 for Adult, MNIST, and CIFAR-10, respectively, based on statistical analysis during training. All experiments are conducted on a 1080 Ti GPU.

Table 1: Comparison of utility ($M - \ell_{ce}$) for each agent trained with CoreFed and FedAvg. We see that $\sum_{i \in [n]} \frac{u_i(\theta')}{u_i(\theta^*)} < n$ holds, where $\theta'$ denotes the weights of shared model trained by FedAvg and $\theta^*$ by CoreFed.

| Dataset | Method | Agent 0 | Agent 1 | Agent 2 | U(Average) | U(Multi) | $\sum_{i \in [n]} \frac{u_i(\theta')}{u_i(\theta^*)}$ |
|---|---|---|---|---|---|---|---|
| Adult | FedAvg | 2.59 | 0.77 | 1.46 | 1.61 | 2.91 | 2.80 (<3) |
| | CoreFed | 2.62 | 0.90 | 1.53 | 1.68 | 3.61 | |
| MNIST | FedAvg | 0.34 | 0.29 | 0.92 | 0.52 | 0.091 | 2.66 (<3) |
| | CoreFed | 0.36 | 0.41 | 0.91 | 0.56 | 0.13 | |
| CIFAR-10 | FedAvg | 0.63 | 1.40 | 0.51 | 0.84 | 0.45 | 2.62 (<3) |
| | CoreFed | 0.73 | 1.35 | 0.71 | 0.93 | 0.70 | |

Table 2: Comparison of utility ($M - \ell_{ce}$) for each agent trained with CoreFed and FedAvg on CIFAR-10 with network VGG-11.

| Method | Agent 0 | Agent 1 | Agent 2 | U(Average) | U(Multi) | $\sum_{i \in [n]} \frac{u_i(\theta')}{u_i(\theta^*)}$ |
|---|---|---|---|---|---|---|
| FedAvg | 0.25 | 3.25 | 3.46 | 2.35 | 2.89 | 2.25 (<3) |
| CoreFed | 1.63 | 3.17 | 3.32 | 2.71 | 17.15 | |

### 5.2 Evaluation Results

We demonstrate that our CoreFed distributed training protocol in Algorithm 1 achieves the core-stable fairness through comparison with FedAvg on different datasets and settings. Concretely, we perform training with FedAvg and our proposed CoreFed, and then validate whether the utility inequality $\sum_{i \in [n]} \frac{u_i(\theta')}{u_i(\theta^*)} < n$ (see *Implications* after Theorem 2) holds under different settings. Here we treat the

Table 3: Comparison of utility ($M - \ell_{ce}$) for each agent trained with CoreFed and FedAvg on CIFAR-10 in the scenario that some agents have data of low quality (i.e., with added Gaussian noise). The variance of added Gaussian noise is 0.0,0.5,1.0 for agent 0,1,2, respectively.

| Method | Agent 0 | Agent 1 | Agent 2 | U(Average) | U(Multi) | $\sum_{i\in[n]} \frac{u_i(\theta')}{u_i(\theta^*)}$ |
|--------|---------|---------|---------|------------|----------|---|
| FedAvg | 3.28 | 3.30 | 1.42 | 2.67 | 15.37 | 2.74 (<3) |
| CoreFed | 3.26 | 3.27 | 1.95 | 2.83 | 20.79 | |

model trained by FedAvg parameterized by $\theta'$, while the model trained by our CoreFed parameterized by $\theta^*$. That is to say, since the model trained by CoreFed achieves core-stable fairness, we expect the model parameterized by $\theta$ would have pareto-optimality. Indeed, results in Section 5.1 suggest that CoreFed achieves core-stable fairness compared with FedAvg while maintaining similar utility. We also report the average and multiplicative utility of the trained global model in "U(Average)" and "U(Multi)" columns. We can see that CoreFed achieves higher overall utilities, especially for the multiplicative case since FedAvg favors the average case in general. We validate the conclusion on more complex DNN VGG-11 on CIFAR-10 in Table 2. In addition, we explicitly consider agents with low quality data by adding Gaussian noises to some agents as shown in Table 3, demonstrating the optimality of CoreFed. We also perform the evaluation with more agents as shown in Table 4.

Table 4: Comparison of utility ($M - \ell_{ce}$) for each agent trained with CoreFed and FedAvg on CIFAR-10 with more agents. $Ai$ represents the $i$-th agent. $U_A$ and $U_M$ denote the average and multiplication of the utility respectively.

| Method | A0 | A1 | A2 | A3 | A4 | A5 | A6 | A7 | A8 | A9 | $U_A$ | $U_M$ | $\sum_i \frac{u_i(\theta')}{u_i(\theta^*)}$ |
|--------|----|----|----|----|----|----|----|----|----|----|-------|-------|---|
| FedAvg | 2.11 | 2.30 | 3.04 | 3.28 | 1.15 | 2.70 | 2.00 | 2.72 | 2.76 | 3.14 | 2.52 | 7084 | 9.77(<10) |
| CoreFed | 2.26 | 2.50 | 3.12 | 3.32 | 1.42 | 2.50 | 1.99 | 2.80 | 2.65 | 2.99 | 2.55 | 9173 | |

## 6 Conclusion

In this work, we introduce a new notion of fairness in federated learning, inspired from a fundamental concept in social choice theory. We show that the new fairness notion is more resilient to noisy data from certain clients, in comparison to FedAvg or egalitarian fair federated learning methods. We believe this work would open up new research directions on connecting game theoretic analysis and statistical machine learning under different learning paradigms, objective, and utilities.

**Acknowledgements** This work is supported by the NSF CAREER grant CCF No.1750436, NSF grant No.1910100, NSF CAREER grant CNS No.2046726, C3 AI, and the Alfred P. Sloan Foundation.

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
