# OpenReview forum: "Fairness in Federated Learning via Core-Stability"
_NeurIPS.cc/2022/Conference — NeurIPS 2022 Accept_

### Official Review · Reviewer_JpwV · 2022-07-03

**Rating:** 6
**Confidence:** 3
**Soundness:** 2 fair
**Presentation:** 2 fair
**Contribution:** 2 fair

**Summary:**

The paper introduces a new notion of fairness dubbed as core-stable fairness in FL setting, by borrowing ideas from game theory and social choice theory. Core-stable fairness indicates there is no subset of agents that can benefit significantly by forming a coalition. The authors propose a new FL Protocol based on this new notion of fairness called CoreFed. The authors prove that CoreFed finds a core-stable model when the loss functions of agents are convex, and CoreFed can find an approximate model when loss functions are non-convex.  Finally the authors do an empirical comparison between CoreFed and FedAvg in which they show CoreFed achieves higher core-stable fairness than FedAvg while maintaining similar accuracy.

**Questions:**

1. What are the $privacy$ implications of Core-stable fairness ? Some discussion around this in the paper would be useful.
2. Similarly, What are the implications of CoreFed protocol, in regards to communication-cost given that in addition to local updates, local losses need to be also communicated back to the server. How scalable this protocol is in real world applications ?

**Strengths And Weaknesses:**

**Strengths:**
* The paper appears to be theoretically sound as well as the claims seem to be supported by theoretical proofs.
* The paper introduces a new notion of fairness in FL settings, that has not been studied in the literature.

**Weaknesses:**

Mainly is due to limited empirical evaluation. In particular:
* The proposed protocol/algo only compared to FedAvg, and not with some existing recent works in the Literature, e.g., Ref 20 [Li et al., 2020].
* Limited experiments and few number of agents (i.e., 3 agent in Table 1 ).
* Lack of experiments in settings with low quality data from some agents. This was a promise of the proposed algorithm to be robust against low quality agents in the summary and main body of the paper but was not supported by experiments.

*Minor Comments (not affecting review score):*

Duplicate references in introduction [32, 23, 8, 35, 36, 26, 23]

---

> ### Author Response · Authors · 2022-08-02
> **Addressing questions of Reviewer JpwV**
>
> Thank you so much for the detailed review and suggestions.
>
> **Question 1:**
> What are the privacy implications of Core-stable fairness? Some discussion around this in the paper would be useful.
>
> **Answer:** This is a great question. To answer this question, we first observe that Core-Fed only differs from Fed-Avg subtly in the way the server aggregates the agent updates.
>
> In particular, in Fed-Avg, in a round $t$, the server updates the current model $\theta^{t-1}$ by simply taking the average of updates ($\Delta_i^{t-1}$ from each agent $i$) from a set $\mathcal C$ of $k$ clients chosen at random, i.e.,  $\theta^t = \theta^{t-1} + \frac{1}{k} \sum_{i \in \mathcal C} \cdot \Delta_i^{t-1}$.  In Core-Fed, the server scales each user-update further by the utility of the agent, i.e.,
>   $\theta^t = \theta^{t-1} + \frac{1}{k} \sum_{i \in \mathcal C}  \cdot \frac{\Delta_i^{t-1}}{u_i(\theta^{t-1})}$.
>
> At a high-level, we believe that since the agent updates are extremely high dimensional vectors, while the agent utility (or equivalently the agent loss) is a scalar, Core-Fed does not involve agents communicating significantly more information than Fed-Avg, and should therefore have similar privacy guarantees as Fed-Avg.
>
> Alternatively, we can re-define the utility of agent $i$ as $u'_i(\theta) = \log(u_i(\theta))$. In this case, observe that the local updates by agent $i$ in Fed-Avg, namely in the gradient of $u'_i()$, will automatically have the scaling by $1/u_i(\theta)$, and thereby the server aggregation of Fed-Avg on the agents with modified utilities will be exactly the same as that of Core-Fed (with Core-Fed run on the original instance with $u_i$s and Fed-Avg run on the new instance with modified utilities $u'_i$s).
>
>
>
> **Question 2:**
> Similarly, What are the implications of CoreFed protocol, in regards to communication-cost given that in addition to local updates, local losses need to be also communicated back to the server. How scalable this protocol is in real world applications?
>
> **Answer:** The reviewer correctly points out a crucial difference in the communication of Core-Fed and Fed-Avg. However, we believe that the increase in communication cost is not quite significant, primarily because local updates are very high dimensional vectors, while the local loss is only a scalar. Therefore, we believe that this one scalar should not add a significant extra cost in communication.
>
> **Response to Weakness 1:** We compare only to FedAvg as Core-Fed protocol is very similar to Fed-Avg with certain subtle but crucial differences. We wanted to show the improvements we can achieve by these subtle changes.
> Nevertheless, we agree that a comparison to other Federated Learning protocols will be interesting and we will do this in the final revision of our paper
>
> **Response to Weakness 2:** We have followed the suggestion to add experiments on more agents (10 agents) and provide the results as below. We can see that with more agents, the proposed CoreFed still achieves higher fairness results than the baseline.
>
> | Protocol                                        | Agent 0     | Agent 1    | Agent 2   |  Agent 3 | Agent 4 |  Agent 5 | Agent 6| Agent 7| Agent 8| Agent 9|  U(Avg)  | U(Multi)    | $\sum_{i \in [n]} \frac{u_i(\theta')}{u_i(\theta^*)}$ |
> |------------------------------------------|-----------|---------|-------|------|-----|----|------|-------|-----|-------|-------------|--------|----------|
> | Fed-Avg  | 2.11     | 2.30          | 3.04    |    3.28  |    1.15  | 2.70   | 2.00 | 2.72  | 2.76 | 3.14  | 2.52 |7084 | 9.77 < 10   |
> | Core-Fed | 2.26     |  2.50          |  3.12 |      3.32   |    1.42  | 2.50    | 1.99  | 2.80  | 2.65   | 2.99  | 2.55  |9173   |   |
>
> **Response to Weakness 3:**   We have followed the suggestion to add experiments on agents with low quality data (Gaussian noise). In particular, we conduct evaluation when the data is distributed uniformly but some agents have data with Gaussian noise. We use the CNN model, and the agents contain data with Gaussian noise (variance $\sigma$=0.0, 0.5, 1.0 for agents 0,1,2). We observe a significant improvement in the utility of the agent with lowest noise, namely Agent 0, in CoreFed compared to FedAvg. This suggests that the proposed method achieves higher fairness than the standard FedAvg.
>
> | Protocol                                        | Agent 0                                     | Agent 1                | Agent 2   | U(Avg)  | U(Multi)    | $\sum_{i \in [n]} \frac{u_i(\theta')}{u_i(\theta^*)}$ |
> |---------------------------------------------|------------------------------------------|-------------------------|-------------|--------|----------|---------|
> | Fed-Avg  | 1.42      | 3.30          | 3.28    |    2.67   |    15.37  | 2.74 < 3   |
> | Core-Fed | 1.95     |  3.27          |  3.26 |      2.83   |    20.79  |     |

---

### Official Review · Reviewer_qZPM · 2022-07-10

**Rating:** 7
**Confidence:** 3
**Soundness:** 3 good
**Presentation:** 3 good
**Contribution:** 3 good

**Summary:**

This paper studies core stability, where no subset of agents can benefit much from forming a new coalition, in federated learning. This paper shows that:
1. Core-stable solution exists when the utilities are continuous and the set of maximizers of any conical combination of the utilities is convex.
2. Maximizing $\sum \log(u_i(\theta))$ derives a core stable solution when $u_i$s are concave. The objective can be maximized efficiently through SGD. Interestingly, the gradient update is similar to FedAvg.
3. The work then extend the core-stability to approximate core-stability in the non-convex case.

**Questions:**

For the definition of core-stability, could the authors explain the intuition/motivation that the multiplicative constant is $\frac{|S|}{n}$? Intuitively, the players in $S$ have incentives to form a new coalition if $u_i(\theta')>u_i(\theta)$ for all $i\in S$ or approximately $u_i(\theta')>(1+\alpha)u_i(\theta)$ for some $\alpha>0$.

Is there any technical challenge of extending the results to the setting where core stability is defined as $\nexists \theta', u_i(\theta')>(1+\alpha)u_i(\theta)$ for all $i\in S$? If so, could the authors list some of them?

**Strengths And Weaknesses:**

The paper studies the core-stability in federated learning, showing the existence of core-stable solutions and providing an efficient algorithm to find the solution. The results are novel and clean. The presentation is clear. The algorithm CoreFed is quite subtle as it is similar to FedAvg only with different weight updates. Overall, it is a good work.

My only concern is about the definition of core-stability, where players are incentivized to form a new coalition only when $\frac{|S|}{n}u_i(\theta')\geq u_i(\theta)$. That is to say, a single player would prefer to learn by herself only if her utility can be increased to $n$ times the currently utility. The multiplier $\frac{|S|}{n}$ seems a little weird to me. It seems like that the results make more sense when $n$ is small.

---

> ### Author Response · Authors · 2022-08-02
> **Addressing questions of Reviewer qZPM**
>
> Thank you so much for your comments and questions.
>
> **Question:**
> For the definition of core-stability, could the authors explain the intuition/motivation that the multiplicative constant is $|S|/n$.
> Intuitively, the players in $S$ have incentives to form a new coalition if $u_i(\theta') > u_i(\theta)$ for all $i \in S$ or approximately $u_i(\theta')> (1+\alpha)u_i(\theta)$ for some $\alpha >0$.
>
> Is there any technical challenge of extending the results to the setting where core stability is defined as there exists no $\theta'$ such that $u_i(\theta') > (1+\alpha)u_i(\theta)$ for all $i \in S$. If so, could the authors list some of them?
>
>
>
> **Answer:**
> We agree that the reviewer's intuition is correct. However, there is an impossibility result for removing the multiplicative factor of $|S|/n$. This is primarily attributed to the heterogeneity of the data. Observe that the agent's utilities are computed based on loss incurred by them on their *own training data*. Consider the scenario when $S = \{i\}$, i.e., $S$ comprises of only a single agent $i$. Without the multiplicative scaling, one would expect that $u_i(\theta') \leq u_i(\theta)$ for all $\theta'$, i.e., agent $i$ gets the best possible utility (least possible loss). Since our claim should hold for all values of $S$, it implies that each agent gets their best possible utility (or least possible loss) when we choose $\theta$ as the predictor. This is not possible if agents have heterogeneous data, since the best predictor for one agent may give high loss values on the data of another agent.
> However, we agree that proving improved multiplicative guarantees for Core-Fed in a "limited heterogeneity" setting is a very interesting avenue for future research.
>
>
> **Weaknesses:**
> My only concern is about the definition of core-stability, where players are incentivized to form a new coalition only when $\frac{\lvert S \rvert}{n} u_i(\theta') \geq u_i(\theta)$. That is to say, a single player would prefer to learn by herself only if her utility can be increased to times the currently utility. The multiplier $|S|/n$ seems a little weird to me. It seems like that the results make more sense when is small.
>
> **Response:**
> Intuitively, it is saying that, if a subset $S$ of agents train their own classifier, then it may do well on their own training data, but no more than by a factor of $\frac{n}{|S|}$. The multiplier of $\frac{|S|}{n}$ is unavoidable due to impossibility results discussed above.
>
> We note that since the guarantee holds for all subsets $S$, it gives us Proportionality and Pareto-optimality at the same time as a byproduct, which itself is hard to achieve.

---

> > ### Comment · Reviewer_qZPM · 2022-08-08
> > **Thanks for your response**
> >
> > Thanks for your answer.
> >
> > I agree that a large multiplicative factor is unavoidable in the heterogeneous setting. Can you elaborate more on why the specific choice of $\frac{|S|}{n}$ is unavoidable? Could you provide a toy example?
> >
> > Consider the binary classification task (0-1 loss) and $u_j(\theta) = 1+\epsilon - 0.9 =0.1+\epsilon$, which means $\theta$ is a really bad solution for agent j. But $\theta$ is still core-stable if $n\geq 10$. Intuitively, agent j definitely has incentives to leave this coalition as even random prediction can give the agent a better utility value.

---

> > > ### Author Response · Authors · 2022-08-09
> > > **Thank you for the response and request for clarification**
> > >
> > > Thank you very much for going through our response. We first give an explicit example showing that the multiplicative guarantee of  $|S|/n$ is tight. Then, we address the concerns raised by the example provided by the reviewer.
> > >
> > > **Example to show that $|S|/n$ is tight:**  Our guarantees work under the assumption that the utility functions are concave and the domain of predictor is a convex set. Consider the following scenario: Our goal is to choose a predictor $c \in \mathcal C$, where $\mathcal C = \{c \in \mathbb{R}^n_{\geq 0} \mid  \sum_{i \in [n]} c_i =1 \}$ (so $\mathcal C$ is a convex set). Now, let there be $n$ agents, and $u_i(c) = c_i$ (utility functions are linear and therefore concave). Intuitively, each agent has their ideal predictor to be a distinct axis aligned hyperplane (capturing the heterogenity in data). Observe that for each agent $i$, the best possible utility is $1$, as there exists a predictor $c^*$ such that $c^*_i=1$ and $c^*_k = 0$ for all $k \neq i$.
> > >
> > > Now, note that, for any predictor $c \in  \mathcal C$, we have $\sum_{i \in [n]} u_i(c) = \sum_{i \in [n]} c_i =1$. Thus, for any classifier $c$ chosen, there exists an $i$, such that $c_i \le 1/n$ and thereby for agent $i$, $u_i(c) = c_i \leq 1/n$. Thus, for each predictor $c \in \mathcal C$, there exists a set $S = \{i\}$ and a predictor $c^* \in \mathcal C$, such that  $\frac{u_i(c)}{u_i(c^*)} = 1/n = |S|/n$. Observe that even our guarantees of proportionality are tight in this example. This example shows that the factor of $|S|/n$ is unavoidable.
> > >
> > >
> > > **The example pointed out by the reviewer:** We would like to point out two subtleties to the classifier being core-stable. Firstly, the example provided by the reviewer is indeed proportional if $n \geq 10$, but for core-stability, we need similar guarantees on every possible set $S$, i.e., for all subsets $S \subseteq [n]$, there exists no agent $i \in S$ and a classifier $c^*$, such that $\frac{|S|}{n} \cdot u_i(c^*) \geq u_i(c)$ for all $i \in [n]$. Note that this implies that for all subsets of size $n/2$, there exists no classifier that can give every agent in the subset twice the utility given by the core-stable classifer. For larger subsets, the utilitarian guarantees provided by a random classifier can be significantly worse than that provided by a core-stable classifier.
> > >
> > > Secondly, our guarantees hold for the worst-case instances (like the one shown above). In practice, Core-Fed may find a classifier with guarantees better than the theoretical guarantees that we provide, as the agent's data may not be that heterogeneous. A systematic study of the same can be done by formalizing the notion of heterogeneity of data across the agents and then give approximation guarantees for core-stability, parameterized by the heterogeneity.
> > >
> > > Thank you again for an insightful ques
> > > tion. We will try to add a discussion to this end in the  final version of our paper.

---

> > > > ### Comment · Reviewer_qZPM · 2022-08-09
> > > > **Thanks**
> > > >
> > > > Thanks for your explanation! Your example makes sense. My concern has been addressed.

---

### Official Review · Reviewer_WXrM · 2022-07-11

**Rating:** 7
**Confidence:** 2
**Soundness:** 4 excellent
**Presentation:** 3 good
**Contribution:** 3 good

**Summary:**

This paper introduces a definition for fairness between agents in federated learning that is derived from social choice theory. The authors name it core-stability. Intuitively, core-stability ensures that there is no incentive for any subgroup $S$ of agents to defect from the rest and create their own centralized model. The authors point out how core stability ensures that two basic conditions of proportionality, and Pareto optimality are satisfied. Moreover, the authors offer an optimization objective CoreFed which provably converges to a core-stable network under a set of reasonable assumptions (the definition of core-stability can be slightly relaxed to be local in order for the theory to extend more diverse settings). Finally, the authors provide some empirical results to show that CoreFed does better than the existing FedAvg objective at achieving core-stable fairness.

**Questions:**

I have decided to accept this paper. I would raise my score even further if the authors could better contextualize their work to offer an explanation of perhaps why this notion of core-stability (which is quite natural) has not been used in the past, and the impact they believe it will have now.

**Ethics Review Area:**

["I don’t know"]

**Limitations:**

Yes.

**Strengths And Weaknesses:**

## Originality
---
### Strengths
- Though I am not familiar with the literature on the intersection of fair federated learning schemes and social choice theory, I was unable to find any work that used this notion of core-stability. As far as I can tell, this goal of essentially subgroup indifference has not been proposed before in federated learning. Moreover, I believe this is a fresh perspective that should be further explored in future work.

## Quality and Clarity
---
### Weaknesses
- Some of the writing in the paper was needlessly wordy, particularly in the introduction and the abstract, and there were some minor grammatical issues throughout the paper. I think for the camera ready version it would be good to trim down these sections and polish the prose a bit further, but these problems do not get in the way of understanding the paper.
- The authors could have performed more empirical results to further investigate CoreFed on more commonly used and complex models than the simple CNN used for section 5.2. More comprehensive empirical results in general would be a great avenue for future work.

### Strengths
- The quality of the theoretical analysis in this paper is exceptional. The authors demonstrate that they have thought deeply about this problem from many different angles, and the math in the main body was all easy to follow (intuition for omitted proofs was quite good too). Though I didn't carefully read the proofs in the appendix, everything looked sound at a glance.

## Significance
---
It's difficult for me to assess the significance of this work in the absence of experience in this line of research. What I can say is the claims are ostensibly sound, and the idea is very well motivated.

---

> ### Author Response · Authors · 2022-08-02
> **Addressing questions of reviewer WXrM**
>
> Thank you so much for your comments, questions and suggestions.
>
> **Question:**
> I have decided to accept this paper. I would raise my score even further if the authors could better contextualize their work to offer an explanation of perhaps why this notion of core-stability (which is quite natural) has not been used in the past, and the impact they believe it will have now.
>
> **Answer:**
> This is indeed a great question and we agree that adding a remark/subsection on this will help improve the exposition of the paper.
>
> Core-stability is a fundamental notion in game theory and computational social choice, formally defined by Gillies in 1959. The reason it has not been used within the federated learning (FL) setting so far may be two fold: (i) The trend of importing notions from computational social choice to fairness in federated learning, or in ML algorithms in general, is relatively recent. For example, only recently an egalitarian fairness notion was adapted to federated learning by Donahue and Kleinberg [1]. (ii) Core-stability is not directly applicable when agent's preferences are defined with respect to losses (or costs). Even if one is able to capture losses through a utility function, like we did, core-stability is known not to exist in many well studied settings. (for instance zero-sum games), especially when preferences are not ``nice'' like convex/concave, which is the case in FL when agents use DNN to train.
>
> For these reasons, we believe, we may be the first to define core-stability in the federated learning setting, and thereby manage to show exact/approximate guarantees for the well-studied classification methods including DNN under some conditions. In addition to providing an efficient federated learning protocol with important fairness guarantees, we hope our paper will instigate more such transfers of concepts between machine learning and computational social choice.
>
> **Weakness 1:** Some of the writing in the paper was needlessly wordy, particularly in the introduction and the abstract, and there were some minor grammatical issues throughout the paper. I think for the camera ready version it would be good to trim down these sections and polish the prose a bit further, but these problems do not get in the way of understanding the paper.
>
> **Response:** Thank you for pointing it out. We will re-write some part of the introduction, and correct all the grammatical errors that we spot in the next revision.
>
>
> **Weakness 2:** The authors could have performed more empirical results to further investigate CoreFed on more commonly used and complex models than the simple CNN used for section 5.2. More comprehensive empirical results in general would be a great avenue for future work.
>
>
> **Response:** Thanks for the suggestion. We have conducted evaluations on more complex model following the suggestions using VGG-11 on CIfar-10. The results are shown below and we can see that our proposed Core-Fed achieves higher fairness than the baseline.
>
>
> | Protocol                                                     | Agent 0                                      | Agent 1               | Agent 2          |U(Avg)            |U(Multi)   |  $\sum_{i \in [n]} \frac{u_i(\theta')}{u_i(\theta^*)}$  |
> |---------------------------------------------|------------------------------------------|-------------------------|-------------------|-------------------|-------------|--------------|
> | Fed-Avg       |   0.25         | 3.25    |  3.46    |  2.35   |     2.89|     2.25 < 3|
> | Core-Fed      |  1.63         | 3.17     | 3.32    |  2.71   |   17.15|                   |
>
> Table: Comparison of Fed-Avg and Core-Fed. $\theta'$ is the predictor trained by Fed-Avg and $\theta^*$ is the predictor trained by Core-Fed.
>
>
> [1] Kate Donahue and Jon M Kleinberg. \textit{Models of Fairness in Federated Learning.}  CoRR, abs/2112.00818, 2021.

---

> > ### Author Response · Authors · 2022-08-06
> > **Follow-up discussion**
> >
> > We thank the reviewer for the valuable questions and suggestions. We hope our answers were able to address the questions, and please let us know if you have other questions.
> >
> > We look forward to follow-up discussions to further improve our work. Thank you!

---

> > ### Comment · Reviewer_WXrM · 2022-08-07
> > **Response to Author Rebuttal**
> >
> > Thank you for the discussion of the context of social choice theory, and the additional experiments. I hope these things are somehow fit into the camera ready version, as I believe they make the paper more compelling. I have raised my score to a 7.

---

### Author Response · Authors · 2022-08-02
**Overall Comment to the Reviews**

We thank the reviewers for their time, suggestions and great questions that we strongly believe can improve the quality of the paper. Below, we summarize our overall response to the reviewer's comments and questions.

 1.  We have explained the possible reasons for why core-stability has not been explored so far for federated-learning, and the possible impacts of our paper, in the response to reviewer WXrM.

 2.  We have provided the intuition and necessity behind the multiplier $|S|/n$ in the definition of core-stability, in the response to reviewer qZPM.

 3. We have discussed possible privacy and communication cost of CoreFed compared to the standard Fed-Avg, which we believe is very low, in the response to reviewer JpwV.

 4. We have added new experimental results: $(i)$ with more complex model than CNN, namely VGG-11 on CIfar-10, $(ii)$ with more agents, $(iii)$ with agents that have very poor quality data. The table corresponding to $(i)$ is in the response to reviewer WXrM, and those of $(ii)$ and $(iii)$ are in the response to reviewer JpwV.

---

### Meta-Review · Area_Chair_B2tx · 2022-08-25

**Recommendation:** Accept
**Confidence:** Certain

**Metareview:**

This paper introduces core stability as a fairness notion for federated learning, which is motivated by social choice theory. The reviewers all agreed that the paper provides a novel contribution to studying fairness in federated learning. The authors have also addressed reviewers' questions during the discussion period.

**Award:**

No

---

### Decision · Program_Chairs · 2022-09-14

Accept